# Cholecystectomy Significantly Alters Gut Microbiota Homeostasis and Metabolic Profiles: A Cross-Sectional Study

**DOI:** 10.3390/nu15204399

**Published:** 2023-10-17

**Authors:** Fusheng Xu, Ruimin Chen, Chengcheng Zhang, Hao Wang, Zhijie Ding, Leilei Yu, Fengwei Tian, Wei Chen, Yongping Zhou, Qixiao Zhai

**Affiliations:** 1State Key Laboratory of Food Science and Resources, Jiangnan University, Wuxi 214122, China; 6210112164@stu.jiangnan.edu.cn (F.X.); 6210113005@stu.jiangnan.edu.cn (R.C.); chengcheng_zhang@sina.cn (C.Z.); edyulei@126.com (L.Y.); fwtian@jiangnan.edu.cn (F.T.); chenwei66@jiangnan.edu.cn (W.C.); 2School of Food Science and Technology, Jiangnan University, Wuxi 214122, China; 3Department of Hepatobiliary, Wuxi No. 2 People’s Hospital, Jiangnan University Medical Center, Wuxi 214002, China; wangyenn@sina.cn (H.W.); dingzhijie@njmu.edu.cn (Z.D.); 4National Engineering Research Center for Functional Food, Jiangnan University, Wuxi 214122, China

**Keywords:** cholecystectomy, gut microbiota, bile acids, diet, nutrient, inflammation

## Abstract

Cholecystectomy (CCE) is a standard clinical treatment for conditions like gallstones and cholecystitis. However, its link to post-CCE syndrome, colorectal cancer, and nonalcoholic fatty liver disease has raised concerns. Additionally, studies have demonstrated the disruptive effects of CCE on gut microbiota homeostasis and bile acid (BA) metabolism. Considering the role of gut microbiota in regulating host metabolic and immune pathways, the use of dietary and probiotic intervention strategies to maintain a stable gut ecosystem after CCE could potentially reduce associated disease risks. Inter-study variations have made it challenging to identify consistent gut microbiota patterns after CCE, a prerequisite for targeted interventions. In this study, we first meta-analyzed 218 raw 16S rRNA gene sequencing datasets to determine consistent patterns of structural and functional changes in the gut microbiota after CCE. Our results revealed significant alterations in the gut microbiota’s structure and function due to CCE. Furthermore, we identified characteristic gut microbiota changes associated with CCE by constructing a random model classifier. In the validation cohort, this classifier achieved an area under the receiver operating characteristic curve (AUC) of 0.713 and 0.683 when distinguishing between the microbiota of the CCE and healthy groups at the family and genus levels, respectively. Further, fecal metabolomics analysis demonstrated that CCE also substantially modified the metabolic profile, including decreased fecal short-chain fatty acid levels and disrupted BA metabolism. Importantly, dietary patterns, particularly excessive fat and total energy intake, influenced gut microbiota and metabolic profile changes post-CCE. These dietary habits were associated with further enrichment of the microbiota related to BA metabolism and increased levels of intestinal inflammation after CCE. In conclusion, our study identified specific alterations in gut microbiota homeostasis and metabolic profiles associated with CCE. It also revealed a potential link between dietary patterns and gut microbiota changes following CCE. Our study provides a theoretical basis for modulating gut microbiota homeostasis after CCE using long-term dietary strategies and probiotic interventions.

## 1. Introduction

The gallbladder (GB) is a pivotal digestive organ in the body, and its primary function is to store and concentrate bile secreted from the liver. During digestion, bile salts are mainly responsible for breaking down fats into particles for easy digestion and promoting the intestinal absorption of fatty acids and fat-soluble vitamins [1]. Cholecystectomy (CCE) is the gold standard treatment for cholecystitis and gallstones [2]. However, the associated risks after CCE are a widespread concern. CCE leads to the unrhythmic secretion of bile, which affects enterohepatic circulation, causes metabolic abnormalities, and predisposes patients to post-cholecystectomy syndrome, which exhibits symptoms such as abdominal pain, diarrhea, and vomiting [3]. Based on epidemiological investigations, CCE is associated with the development of nonalcoholic fatty liver disease (NAFLD) [4], colorectal cancer (CRC) [5], and cardiovascular disease [6], which may be related to the disruption of homeostasis of the gut microbiota and continuous exposure of the intestine to BA.

The gut microbiota is associated with the host immune response and metabolic regulation, and homeostasis is a key factor in maintaining host health [7]. Based on results obtained through epidemiological, physiological, animal, and clinical trials, the risk to human health and disease occurrence may be mediated or altered by microbial communities that live in symbiosis with their hosts [8]. For example, disruption of gut microbiota homeostasis has been shown to be associated with the risk of a variety of diseases, including inflammatory bowel disease (IBD), type 2 diabetes (T2D), NAFLD, CRC, and malnutrition [9]. Disruption of gut microbiota homeostasis is caused by an imbalance between commensal and pathogenic bacteria [10]. In addition, the ecological dysregulation of the gut microbiota is often accompanied by metabolic dysfunction. Therefore, the stability of gut microbiota homeostasis is essential for maintaining normal physiological and metabolic pathways in the host and is a key factor in preventing the development of related diseases.

Current research indicates that CCE can disrupt gut microbiota homeostasis and alter metabolic characteristics [11,12]. This alteration is accompanied by abnormalities in BA metabolism, which is a potential risk factor for the development of related diseases. However, the results of existing studies are often inconsistent and there is still a lack of consensus on CCE-related taxa. In most studies, the adverse effects of increased abundance of the conditionally pathogenic bacteria *Prevotella* [13,14,15] and *Escherichia coli* [16,17], and decreased abundance of the important producers of short-chain fatty acids (SCFAs) *Faecalibacterium* [16] and *Roseburia* [18], have been shown to result from continued exposure of bile to the intestine after CCE, triggering metabolic abnormalities. However, some studies have reported contradictory results. A decrease in the abundance of Firmicutes following CCE was observed in most studies [15,19,20]; however, an increase in its abundance of Firmicutes was observed in a study by Wei et al. [21]. Similarly, the results for *Bacteroidetes* are conflicting. Some studies have shown a decrease in the abundance of these taxa in cholecystectomized populations [14,20] compared to that in controls, whereas others have reported an increase or no change in abundance [15].

Dietary patterns have been extensively studied as key factors affecting health. The risk for CCE is often associated with different dietary patterns. A low-fat-, high-fiber-based diet has been shown to reduce the risk of post-cholecystectomy syndrome [22,23]. The balance of dietary nutrients not only affects the normal physiological metabolic pathway of the host but is also a key factor in maintaining the stability of the gut microbiota. For example, some studies have shown that high fat consumption leads to excessive production of circulating free fatty acids and elevated levels of systemic inflammation [24]. In addition, low-fiber, high-fat, and high-sugar dietary patterns have been associated with alterations in microbiota composition and function and the subsequent development of chronic diseases [25]. Because diet is a key factor in shaping the gut microbiota, the gut microbiota is also often thought of as the medium through which food exerts its anti-inflammatory and pro-inflammatory effects. Therefore, shaping a stable intestinal microecological environment through long-term dietary strategies is an effective means of preventing disease [26].

Inconsistencies between studies may result from differences in the study design and analytical methods used to analyze 16S gene amplicon data, as well as from differences in dietary patterns and lifestyles within the cohort. Therefore, it is difficult to determine significant changes in the gut microbiota after CCE, which is a prerequisite for the long-term use of microbiota as a dietary strategy and a target for probiotic interventions to maintain gut microbiota homeostasis. In our study, we first identified consistent changes in the gut microbiota composition and function following CCE using a meta-analysis. Metabolic characteristics after CCE were characterized in a lab-recruited cohort. Briefly, we used the same pipeline to re-analyze four 16S rRNA original sequencing datasets. We identified the significantly altered gut microbiota after CCE by building a random forest model classifier and applying the bioinformatics tool PICRUSt2 to predict altered gut microbiota functions. Differential gut microbiota were then analyzed for correlations with dietary patterns using dietary information from the food frequency questionnaire (FFQ). In addition, we characterized the specific changes in the fecal metabolome associated with CCE in laboratory-recruited volunteers. Our study demonstrates a significant disruption in gut microbiota homeostasis and fecal metabolic pathways due to varying dietary habits after CCE, as well as an increased risk of intestinal inflammation.

## 2. Materials and Methods

### 2.1. Study Design

#### 2.1.1. Open Database Search

On 18 April 2023, PubMed and Web of Science were used to search for publications containing all keywords “16S,” “gut,” “Cholecystectomy,” the exact phrase “Cholecystectomy,” and at least one of the words “microbiota” [OR] “microbiome” [OR] “gut” [OR] “intestinal” anywhere in the article. This produced 998 results. Subsequently, our search was refined by examining the titles of these search results, and articles that included both the terms “microbiome” (or “microbiota”) and “chole-cystectomy” in their titles were considered for further assessment by reviewing their abstracts.

The inclusion criteria were as follows: 1. The study compared the effects of population-based CCE on the intestinal microbiota; 2. The study had publicly available fecal raw 16S amplicon sequencing data with clearly defined subgroup information, allowing differentiation between the CCE and healthy control (HC) groups; 3. The study methods used were consistent among all selected articles. However, at the time of the search, samples from different cohort design methods or geographical locations were not excluded. Among these, cohort design methods primarily included cross-sectional studies.

#### 2.1.2. CCE Volunteer Recruitment

A total of 34 volunteers were recruited from the Wuxi Second People’s Hospital of Jiangsu Province, comprising 20 individuals in the CCE group and 14 HCs. The clinical data collection and fecal sampling involved in the population trial were carried out in accordance with the protocol approved by the local ethics committee (Registration number: ChiCTR2300073303).

#### 2.1.3. Inclusion and Exclusion Criteria

Patients in the CCE group were selected based on the following criteria: (1) CCE performed more than 2 years ago; (2) age between 18 and 65 years; and (3) provision of written informed consent. Patients were excluded if they had (1) a surgical history of gastrointestinal tract procedures; (2) a medical history of irritable bowel syndrome, IBD, constipation, or infective or idiopathic diarrhea; (3) a medication history of antibiotics, probiotics, or drugs known to affect gut microbiota within the past month; or (4) a history of severe chronic diseases.

HCs voluntarily participated in the study and were selected based on the following criteria: (1) age between 18 and 65 years; (2) provision of written informed consent; (3) no history of GB removal surgery or other gastrointestinal surgeries; and (4) no administration of antibiotics or probiotics in the month preceding the study.

Fresh fecal samples was stored in a laboratory freezer at –80 °C [27].

### 2.2. Dietary Assessment and Questionnaire Processing

A validated semi-quantitative FFQ was used to assess the volunteers’ dietary intake [28]. The average daily nutrient intake was calculated by multiplying frequencies of consumption by portion size and nutrient content per gram [26], as indicated in the China Food Composition Tables [29]. Please refer to the Appendix A for details of the FFQ.

### 2.3. Untargeted Metabolomics

For fecal metabolomic analysis, metabolites were extracted following the protocol outlined by Zhu et al. [30]. Fecal untargeted metabolomics were analyzed using a UIUI3000 high-performance liquid chromatography (HPLC) system (Thermo Fisher Technologies, Waltham, MA, USA) coupled with a high-resolution Q Active Mass Spectrometer (Thermo Fisher Technologies, MA, USA) for detection. Please refer to the Appendix A for detailed information on sample pre-treatment methods, HPLC–MS analytical parameters, and data analysis procedures.

### 2.4. Determination of Fecal SCFA Contents

Fecal SCFAs were determined using gas chromatography–mass spectrometry (GCMS-QP2010 Ultra system, Shimadzu Corporation, Kyoto, Japan), following the method reported by Qu et al. [31]. Briefly, pre-freeze-dried fecal samples of approximately 50 mg were weighed, re-suspended in NaCl solution, and acidified with 10% sulfuric acid, and then SCFAs were extracted using ether. Please refer to the Appendix A for details.

### 2.5. Determination of BA Levels in Fecal Samples

Target BAs in fecal samples were quantified using liquid chromatography–tandem mass spectrometry [32]. For sample preparation, approximately 50 mg of the sample was pre-freeze-dried in advance. Methanol (100%) was added for grinding and homogenization. A 0.22 μm membrane was used for filtration, and the samples were stored in an injection bottle for LC–MS analysis. Refer to the Appendix A for further details.

Fecal total BA levels were quantified using a commercial BA assay kit (Jiancheng Bioengineering Institute, Nanjing, China) [33].

### 2.6. Determination of Calprotectin Levels in the Fecal Samples

Fecal calprotectin levels were quantified using a human calprotectin protein ELISA kit (Shanghai Enzyme-linked Biotechnology Co., Ltd., Shanghai, China).

### 2.7. Data Sources

Raw 16S-rRNA data used in the meta-analysis included data downloaded from public databases (NCBI) and data generated in the laboratory. Detailed fecal gut microbiota genome extraction methods are provided in the Appendix A.

### 2.8. 16S rRNA Data Processing

The bioinformatics analysis process is illustrated in Figure 1. In summary, raw data were analyzed using QIIME2 [34]. The DADA2 package was utilized for quality filtering and demultiplexing of the raw sequencing data. Sequences with a Q-mass fraction less than 20 were filtered. Read segments are assigned using open reference amplicon sequence variation (ASV), and ASV tracking was performed using the Python NumPy and SciPy libraries.

### 2.9. Microbial Community Analysis

GraphPad Prism 10 and R were used for data analysis and visualization. The R software’s R 4.3.1 vegan package was utilized to estimate bacterial community α-diversity (abundance-based coverage estimation (ACE) and Shannon indices) and β-diversity (principal component analysis (PCoA)) indices. Random forest analysis was performed using the online site Wekemo Bioincloud. Receiver operating characteristic (ROC) curves were obtained using the OmicStudio tools. The 16S function was predicted using PICRUSt2, and metabolic pathways were annotated using the Kyoto Encyclopedia of Genes and Genomes (KEGG).

### 2.10. Statistical Analysis

Statistical analyses were carried out using SPSS 25.0 software. Differences in microbiota characteristics, metabolic function, nutrient intake, and metabolite levels were assessed using the Wilcoxon test. The *t*-test was applied to analyze differential fecal untargeted metabolites. Categorical variables were assessed using the χ^2^ test. Statistical significance was defined as a *p* value < 0.05.

## 3. Results

### 3.1. Study Selection

Through our search strategy, 998 studies were identified (Figure 2). However, only three of these studies provided raw sequencing data. Consequently, four studies were included in the analysis, including those with laboratory-generated data (Table 1). These three selected studies comprised a total of 190 samples (97 from the CCE group and 93 from the HC group), with 6 samples lacking annotations. This study incorporated 218 samples (114 from the CCE group and 104 from the HC group). The excluded studies and their details are presented in Appendix A. The characteristics of each cohort are summarized in Table 1 and Table 2. Given the heterogeneity observed among various studies, separate data re-analysis and combinations were conducted.

### 3.2. Microbial Diversity and Composition Comparison between CCE and HC Groups

First, we compared the alpha diversity between the CCE and HC groups using the Shannon and ACE indices. Across all four studies, no significant differences were found between the Shannon and ACE indices (Figure 3A,B). The results of the PCoA showed that the gut microbiota tended to be neostable after CCE (pool, *p* = 0.0004), suggesting that the gut microbiota of the CCE and HC groups tended to have different structures (Figure 3C). There were no significant differences between the two studies (*p* = 0.559 and *p* = 0.788, respectively). To further explore the changes in the intestinal microbiota composition between the CCE and HC groups, we identified differential classes (at the phylum, family, and genus levels) using qualitative analysis. At the phylum level (Figure 3D), the CCE group had a higher relative proportion of *Bacteroidetes* than the HC group (three of the four studies). At the family level (Figure 3D), Veillonellaceae and Lactobacillaceae were more abundant in the CCE group (three of the four studies), whereas Ruminococcaceae exhibited a decreasing trend (three of the four studies). At the genus level (Figure 3D), *Escherichia Shigella*, *Prevotella 9*, *Blautia*, *Ruminococcus gnavus group*, *Megamonas*, *Ruminococcus 2*, and *Lachnoclostridium* were more abundant in the CCE group (three of the four studies), whereas *Faecalibacterium*, *Subdoligranulum*, *Roseburia*, *Ruminococcus torques group*, and *Dialister* shows a downward trend (three of the four studies).

### 3.3. Taxa with Significantly Different Abundances between CCE and HC Groups

Given the notable heterogeneity observed in the gut microbiota across the four cohorts, separate tests were conducted at the family and genus levels for each cohort. This approach allowed consistent alterations in the gut microbiota to be discerned by aggregating data from the CCE and HC groups across all four cohorts. We filtered for species with prevalence and relative abundances of <20% and <0.01% in the raw data, respectively, to ensure that, as far as possible, distinct species were present in each cohort. The results showed consistent alterations in the gut microbiota despite opposing results in different cohorts. At the family level (Figure 4A), CCE significantly increased the abundance of Akkermansiaceae (*p* = 0.018), Lactobacillaceae (*p* < 0.0001), and Muribaculaceae (*p* < 0.0001). At the genus level (Figure 4B), *Akkermansia*, *Erysipelatoclostridium*, *Lachnoclostridium*, *Lactobacillus*, and *Megamonas* were significantly enriched in the CCE group (*p* = 0.019, *p* = 0.012, *p* = 0.048, *p* = 0.0001, and *p* < 0.0001, respectively) and significantly lower abundances of *Collinsella* and *Faecalibacterium* were observed (*p* = 0.026 and *p* = 0.012, respectively) (Figure 4B).

To further determine the significant effects of CCE on the gut microbiota, we distinguished significantly altered gut microbiota characteristics in the CCE group at both the family and genus levels by building a random forest model. The random forest model identified distinct microbiota that distinguished between the CCE and HC groups. At the family level, these mainly included Akkermansiaceae, Bacteroidaceae, Coriobacteriaceae, Eggerthellaceae, and Muribaculaceae. At the genus level, the bacteria mainly consisted of *Bacteroides*, *Megamonas*, and *Acinetobacter*. The discriminatory effect of the random forest model was tested in the validation cohort (*n* = 42 for CCE and *n* = 38 for HC). At the family level, the areas under the curve (AUC) were 0.6869 (Figure 4C) in the discovery cohort and 0.713 (Figure 4D) in the validation cohort. At the genus level, the AUC were 0.6942 (Figure 4E) in the discovery cohort and 0.683 (Figure 4F) in the validation cohort. The ROC curve results showed that the random forest model had medium discriminating ability.

### 3.4. Functional Prediction of Phylogenetic Investigation of Communities by Reconstruction of Unobserved States (PICRUSt2) Analysis between the CCE and HC Groups

Further understanding of the metabolic functions associated with the gut microbiota, we conducted a KEGG pathway analysis to assess potential functional changes corresponding to taxonomic variations. Our analysis integrated data from all four studies for functional prediction. A total of 1312 pathways were identified in both the CCE and HC groups (Figure 5A). In addition, 15 enrichment pathways showed significant differences (*p* < 0.05), including phosphonate and phosphinate metabolism, pyruvate metabolism, butanoate metabolism, the citrate cycle (tricarboxylic acid [TCA] cycle), propanoate metabolism, biosynthesis of secondary metabolites, and carbon fixation pathways in prokaryotes (Figure 5B,C).

### 3.5. Alterations in Gut Microbiota Associated with Diets

As diet is a major factor influencing gut microbiota, we further investigated the dietary intake of the CCE and HC groups using the FFQ. Spearman’s correlation analysis revealed that diet can significantly affect the abundance of different gut microbiota. Mussels, molluscs, meat (both red meat and poultry), oil, and fat were positively correlated with altered Akkermansiaceae (Figure 6A,B) and *Akkermansia* (Figure 6C,D). Pickled pickles and sorghum were found to be negatively correlated with altered *Lachnoclostridium* (Figure 6C). Fruits exhibited a negative correlation with altered *Megamonas* (Figure 6C). Tea displayed a positive correlation with *Collinsella* (Figure 6C), whereas corn and its products, total energy, and fiber were strongly negatively correlated with *Collinsella* (Figure 6C,D). Although not statistically significant, there were positive correlations with Lactobacillus for oil, meat (red meat and poultry), and fat (Figure 6C,D).

### 3.6. Alterations in Fecal BA between CCE and HC Groups

We examined the levels of several BAs in the fecal samples. The results showed that CCE significantly altered BA metabolism (Figure 7A–H). Fecal total BA (Figure 7A), bile acids (CA) (Figure 7B), and β-muricholic acid (β-MCA) (Figure 7D) levels were significantly higher in the CCE group compared to the HC group. In addition, there was a trend towards increased chenodeoxycholic acid (CDCA) levels (Figure 7C).

### 3.7. Alterations in Fecal Metabolome between CCE and HC Groups

We further analyzed the effects of CCE on host metabolic levels using fecal metabolomics. The results showed that the altered gut microbiota after CCE significantly affected fecal metabolism levels (Appendix A). Untargeted metabolomic analysis indicated that CCE significantly altered metabolite classification levels (Figure 8A,B and Appendix A). Compared to the HC group, 3-(4-hydroxyphenyl) propionic acid, 3-hydroxybenzoic acid, D-(-)-fructose, leucylproline, palmitoylcarnitine, xanthurenic acid, and α-linolenoyl ethanolamide were significantly enriched in the CCE group, while xanthine, uracil, pantothenic acid, orotic acid, and hydrocinnamic acid were significantly decreased (Figure 8C). Further KEGG metabolic pathway-based enrichment analyses of the differential metabolites revealed that pantothenate and CoA biosynthesis, pyrimidine metabolism, and thiamine metabolism were significantly downregulated in the CCE group (Figure 8D). Analysis of the fecal SCFAs content showed that the levels of different SCFAs were lower in the CCE group than in the HC group (Figure 8F). Butyric acid content was significantly decreased in the CCE group (Figure 8F). In addition, fecal calprotectin (Figure 8E) contents were significantly higher in the CCE group, suggesting that CCE alters BA metabolism and exposes this group to a higher risk of intestinal inflammation.

## 4. Discussion

Gut microbiota–host interactions have a positive effect on the maintenance of normal immune and metabolic pathways in the host [8]. Disruption of gut microbiota homeostasis is the main cause of several metabolic diseases. Among the numerous factors affecting gut microbiota homeostasis, diet is a key factor in reshaping the intestinal microecosystem [35]. Based on epidemiological findings, CCE may be associated with the development of CRC [36]. The identification of significantly altered gut microbiota after CCE is difficult owing to opposing findings. NAFLD [37] and the development of post-cholecystectomy syndrome are positively correlated with dietary habits characterized by high fat and cholesterol [22,23]. Studies have shown that CCE can significantly disrupt the gut microbiota composition and metabolic function, as well as affect BA metabolic pathways [11,12]. In addition, studies have reported that a high-fat and high-cholesterol diet promotes intestinal inflammation by exacerbating gut microbiome dysbiosis and BA disorders in CCE [38]. Given the pivotal role of the gut microbiota in regulating host immunity and metabolism, strategies aimed at modulating gut microbiota balance through long-term dietary or probiotic interventions could potentially mitigate the risk of CCE. However, targeted modulation of the gut microbiota needs to be based on the identification of significantly altered gut microbiota characteristics after CCE. Therefore, our study identified significantly altered microbiota and metabolic profiles after CCE by meta-analysis and further clarified the role of diet in this modulation.

In our study, we sought to elucidate consistent alterations in the gut microbiota associated with CCE across four independent cohorts. Our findings indicate that CCE significantly altered the compositional structure and β-diversity of the gut microbiota (Figure 3C), although α-diversity did not exhibit significant differences (Figure 3A,B). These results are similar to those of most studies [11,14], suggesting that the structure of the gut microbiota after CCE tends towards a new homeostasis. However, the relationship between gut microbiota diversity and health remains controversial [39]. Therefore, further studies are required to address the biological significance of the altered diversity of gut microbiota after CCE.

We further characterized the differences in the composition of the gut microbiota after CCE. The identification of significantly altered gut microbiota after CCE is difficult owing to opposing findings among different studies. Conventionally, different dietary profiles, disease states, sequencing technologies, and data processing techniques are the main reasons for the heterogeneity between studies. Therefore, we used the same data processing procedures to identify consistent patterns of changes in the gut microbiota that were present in all four studies. The results showed that CCE significantly upregulated the abundance of *Akkermansia*, *Erysipelatoclostridium*, *Lachnoclostridium*, *Lactobacillus*, and *Megamonas* (Figure 4B). In contrast, *Collinsella* and *Faecalibacterium* were depleted in the cholecystectomised populations (Figure 4B). Interestingly, *Akkermansia* [40] and *Lactobacillus* [41], as bacteria with positive effects on host health, can be applied to alleviate various types of diseases through a variety of pathways, such as anti-inflammatory and modulation of immune and metabolic levels. However, normal physiological functioning of the gut microbiota depends on a relatively stable community structure and relative abundance. Studies continue to report that overcolonized *Akkermansia* can participate in disease progression by increasing inflammation and affecting intestinal mucus production, destabilizing the intestinal barrier [42,43]. Therefore, the health effects of *Akkermansia* require further study under various conditions. BAs act as effective bactericides [44], essentially limiting the growth of selected bacteria while enriching the growth of other BA-available bacteria. CCE leads to irregular bile secretion, allowing for increased enterohepatic circulation [3], which further exposes BA to the intestine for a prolonged period of time. This is considered to be the main primary reason why CCE remodels the structure of the gut microbiota. The presence of bile salt hydrolase (BSH) activity is a major requirement for the involvement of gut microbiota in BA metabolism, and *Lactobacillius* contain several species of BSH [45,46]. In addition, studies have shown that *Akkermansia* can regulate BA metabolism, which may occur through direct action or through metabolites [47,48]. Thus, enrichment of *Akkermansia* and *Lactobacillius* may be responsible for the altered BA metabolism after CCE. In addition, studies have confirmed that *Megamonas*, *Erysipelatoclostridium* [49], and *Lachnoclostridium* can cause adverse health effects. For instance, Shinichi et al. [50] used metagenomics and metabolomics to reveal the characteristics of the gut microbiota in patients with early CRC and showed that the abundance of *Megamonas* was significantly elevated in 19.2% of the patients (118 of 616). Another study by Liang J.Q. et al. [51] found that *Lachnoclostrium* sp. was significantly enriched in adenoma patients through metagenomic analysis. Conversely, multiple studies have shown that *Collinsella* [52] and *Faecalibacterium* [53] have some anti-inflammatory effects and produce SCFAs that play a positive role in host health. Specifically, *Faecalibacterium* maintains intestinal barrier stability and plays an anti-inflammatory role by producing SCFAs [54].

Random forests are widely used to distinguish differential microbiota and metabolite profiles and to identify key biomarkers. For example, a random forest model has been constructed to differentiate between significant features of altered gut microbiota or metabolites in patients with obesity [55], gout [56], osteoporosis [57], and COVID-19 [58]. In our study, the random forest classifier performed moderately in identifying key altered flora after CCE at both the family and genus levels (Figure 4C–F). Consequently, our findings suggest that *Bacteroides*, *Megamonas*, and *Acinetobacter* may be among the several groups of bacteria significantly altered after CCE.

Changes in the composition and structure of gut microbiota do not necessarily support significant metabolic functions. We used the PICRUSt2 tool to analyze changes in gut microbiota function. The results showed that the function of the gut microbiota after CCE exhibited significant differences in glycolipid and secondary product metabolism. These mainly include the citrate cycle (TCA cycle), fatty acid biosynthesis, fatty acid metabolism, and secondary metabolite biosynthesis. Boursier et al. [59] reported that the metabolic pathways associated with nonalcoholic steatohepatitis and fibrosis mainly include carbohydrate, lipid, and amino acid metabolism. Changes in fatty acid metabolic pathways may be caused by the altered BA metabolism of substances such as cholesterol and fat following CCE [60]. Therefore, our study shows that CCE changes the composition and structure of the gut microbiota, and that the gut microbiota also shows significant differences in glycolipid metabolism and secondary metabolic functions.

Diet has received considerable attention as a key factor influencing health. In the present study, we further elucidated the effects of different dietary habits on the alteration of the gut microbiota after CCE. Among the volunteers recruited, high fat and energy intakes were the main dietary characteristics of the CCE group (Table 2). We noted that the enrichment of *Akkermansia* and *Erysipelatoclostridium* after CCE was associated with excess fat and total energy intake. A ketogenic dietary pattern characterized by a high fat content supports the enrichment of *Akkermansia*. Olson et al. [61] established a ketogenic diet mouse model and elucidated the roles of gut microbiota in regulating seizure susceptibility. Among these, the enrichment of *Akkermansia* was positively correlated with the intervention of the ketogenic diet. Additionally, a high abundance of *Erysipelatoclostridium* has been reported to be associated with high-fat diets or metabolic abnormalities associated with obesity [49,62]. In our study, the differences between *Megamonas* and *Faecalibacterium* in the CCE and HC groups showed similar concordance with dietary characteristics, although the differences were not statistically significant. One potential cause of *Megamonas* enrichment was excessive energy intake [63]. In contrast, depletion of *Faecalibacterium* correlates with high levels of energy and fat [64]. In conclusion, our findings suggest that different dietary patterns following CCE can alter the composition and metabolic function of the gut microbiota. Therefore, in addition to considering the direct effect of diet on the homeostasis of the gut microbiota, further analysis should be performed to further shape the intestinal microecosystem by changing the physiological metabolism of the host. For example, the enrichment of *Akkermansia* after CCE may be caused by an abnormal BA metabolism, which is exacerbated by excessive fat intake.

In our study, we provided evidence that CCE disrupts intestinal homeostasis, leading to discernible metabolic differences. The results showed that fecal calreticulin levels were significantly higher in the cholecystectomised group than in the HC group (Figure 8E), suggesting that CCE may elevate intestinal inflammation levels. Fecal calprotectin is a marker of intestinal inflammation that is significantly elevated in the intestines of patients with IBD [26]. In addition, CCE significantly up-regulated the fecal levels of total BA, CA, and β-MCA metabolism (Figure 7A,B,D). These results confirm that CCE disrupts BA metabolic pathways, potentially leading to prolonged exposure to BA within the intestine. In contrast, five SCFAs showed a decreasing trend in the CCE group (Figure 8F), particularly butyric acid, which was significantly lower than that in the HC group. SCFAs are major products of gut microbiota metabolism, maintain intestinal barrier stability, and improve intestinal anti-inflammatory properties [65]. CCE destroys the major SCFA-producing bacteria, resulting in a decrease in SCFA content. Enrichment analysis of fecal untargeted metabolites, based on the KEGG database, revealed that CCE significantly downregulated pantothenate and CoA biosynthesis, pyrimidine metabolism, and thiamine metabolism. Interestingly, pantothenate and CoA biosynthesis [66] and pyrimidine metabolism [67] are associated with lipid and energy metabolism, and also affect the occurrence of diseases such as inflammation and tumors by regulating immune metabolic pathways [68,69]. In conclusion, we characterized fecal metabolic profiles after CCE. Our results suggest that CCE may increase intestinal inflammation by altering gut microbiota homeostasis and host metabolic profiles.

Our study had several limitations. Firstly, the risk associated with post-CCE accumulates over time, but we lacked precise information on the time elapsed since CCE for each sample in the cohort. Therefore, we could not assess the impact of CCE duration on gut microbiota homeostasis and metabolic characteristics. Secondly, the interaction between diet and gut microbiota is complex, and we primarily focused on identifying associations between differential microbiota and dietary factors after CCE without delving into the underlying causes and potential physiological implications of this interaction. Thirdly, we used only two databases for the search, which may have resulted in missing data. Lastly, since the published studies provided only 16S data, we lacked insights into how CCE specifically alters the functional genes of the gut microbiota, such as the expression of genes encoding enzymes related to BA metabolism. For future investigations, we aspire to gain a deeper understanding of the risks linked to post-CCE and the factors that can mitigate these risks through comprehensive epidemiological studies in larger cohorts. In addition, an in-depth analysis of the alteration of gut microbiota homeostasis and related functional genes by different dietary patterns after CCE can guide us to better understand the mechanism underlying the effect of CCE on intestinal health.

## 5. Conclusions

Our study revealed that CCE significantly altered gut microbiota homeostasis and associated metabolic profiles. On this basis, we further demonstrated the role of different dietary patterns in remodeling gut microbiota homeostasis after CCE. Enrichment of microbiota associated with BA metabolism and altered function associated with lipid metabolism are key features accompanied by elevated fecal calprotectin and total BA levels and decreased SCFA content. Collectively, these findings imply a heightened risk of intestinal inflammation associated with CCE and underscore the pivotal role played by the disruption of gut microbiota equilibrium and metabolic profiles in the development of intestinal inflammation. This study presents a fresh perspective for assessing the risks following CCE, moving beyond mere correlations and delving into potential causative factors. Furthermore, it lays the foundation for targeted interventions aimed at enhancing post-CCE intestinal health. Such interventions may involve long-term dietary strategies and prebiotic interventions, harnessing the power of gut microbiota modulation to mitigate post-CCE risks and improve patient outcomes.

## Figures and Tables

**Figure 1 nutrients-15-04399-f001:**
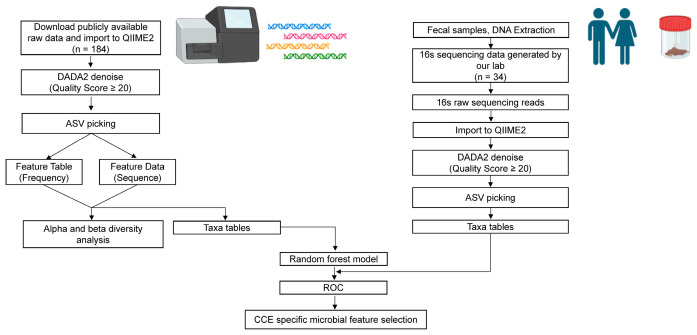
Bioinformatics analysis pipeline.

**Figure 2 nutrients-15-04399-f002:**
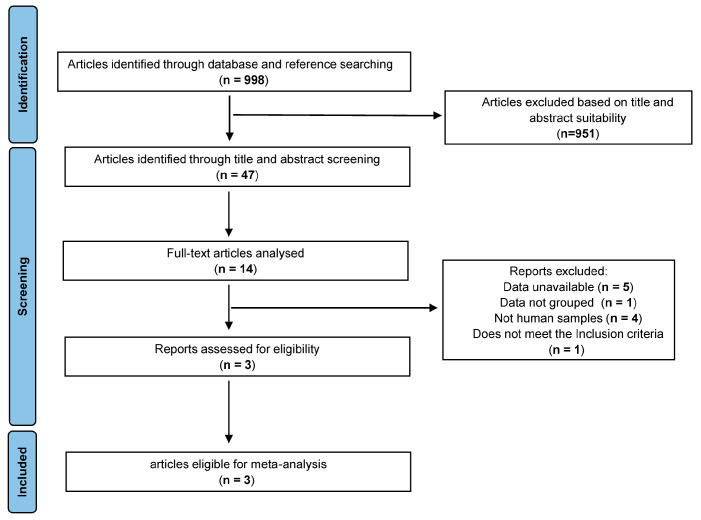
Flowchart illustrating the search strategy for the meta-analysis.

**Figure 3 nutrients-15-04399-f003:**
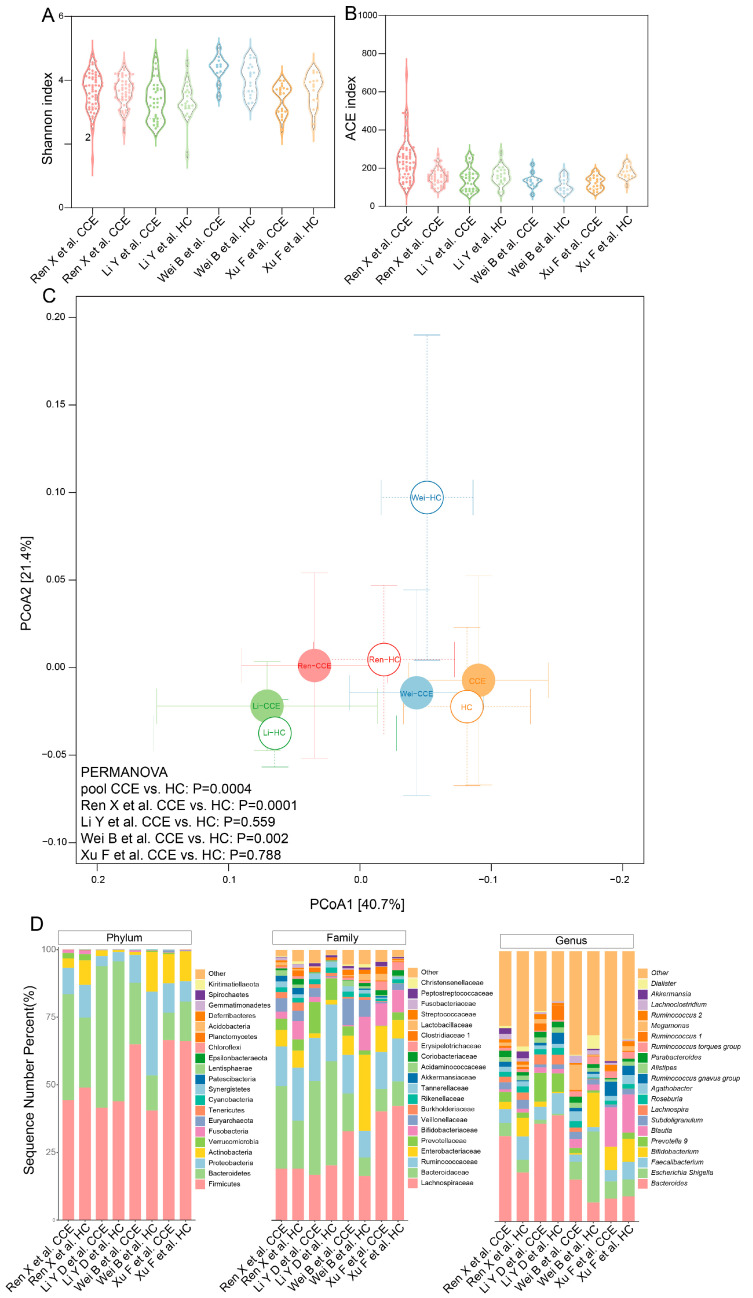
Effects of CCE vs. HC on gut microbiota diversity. (**A**,**B**) Shannon (**A**) and ACE (**B**) indices between CCE and HC groups from each study. Data are expressed in a violin plot as min to max. (**C**) PCoA analysis of Bray–Curtis dissimilarities obtained for the 16S data in the CCE (closed circles) and HC (open circles) groups using permutational multivariate analysis of variance. (**D**) Relative abundance of species in a cylindrical accumulative graph of bacteria in the CCE and HC groups.

**Figure 4 nutrients-15-04399-f004:**
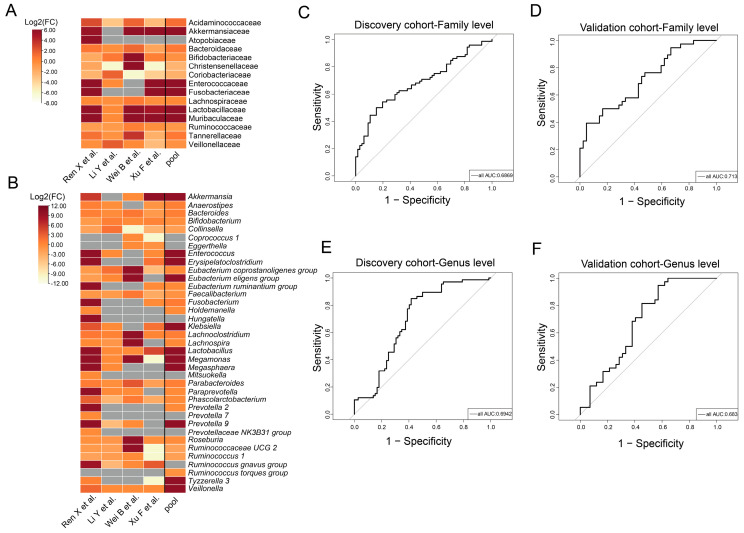
Taxonomic differences were detected between the CCE and HC groups. (**A**) Differences in gut microbiota at the family level between the CCE and HC groups in each study. Data in the heat map are presented using log_2_-fold change [FC]. (**C**,**D**) ROC curves were used to evaluate the ability of the random forest and to identify the gut microbiota at the family level that were significantly changed in the CCE group. Performance of the classifiers in the discovery (**C**) and validation (**D**) cohorts, as assessed using the AUC. (**B**) Differences in gut microbiota at the genus level between the CCE and HC groups in each study. Data in the heat map are presented using log_2_-(FC). (**E**,**F**) ROC was used to evaluate the ability of the random forest to identify the gut microbiota at the genus level that was significantly changed in the CCE group. Performance of the classifiers in the discovery (**E**) and validation (**F**) cohorts, as assessed using the AUC.

**Figure 5 nutrients-15-04399-f005:**
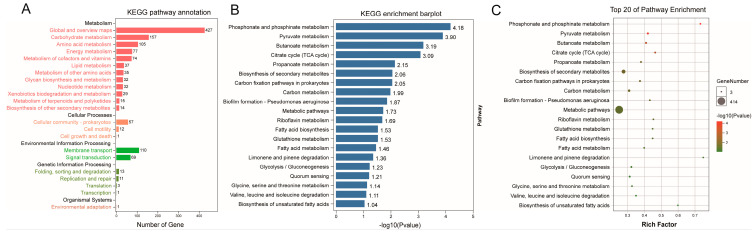
The prediction of metabolic function within the microbiome in the CCE and HC groups. (**A**) Enrichment histogram of differential KEGG metabolic pathways in the CCE group. The numbers on the bar chart represent the number of KEGG B classes. (**B**) Statistical difference histograms of metabolic pathways. (**C**) Rich factor reflects the proportion of differential genes in the KEGG metabolic pathways to the total number of genes affecting each pathway. Bubble size and color reflect gene number and significance, respectively.

**Figure 6 nutrients-15-04399-f006:**
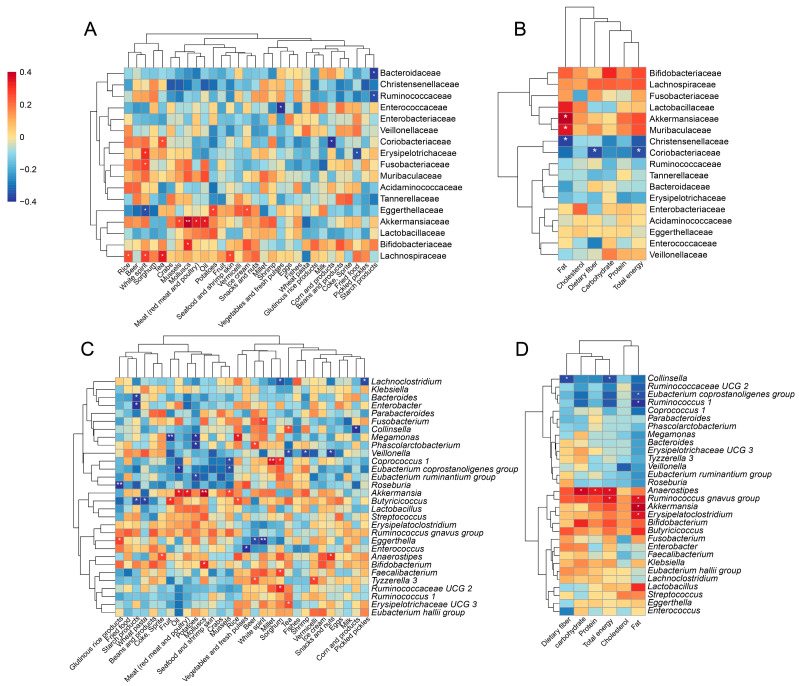
Alterations in gut microbiota after CCE influenced by diets. (**A**,**C**) Analysis of the correlation between gut microbiota (at the family level) abundance and dietary intake (**A**) and nutrient intake (**C**) using Spearman’s correlation. (**B**,**D**) Analysis of the correlation between gut microbiota (at genus level) abundance and dietary intake (**B**) and nutrient intake (**D**) using Spearman’s correlation. * *p* ≤ 0.05, ** *p* ≤ 0.01.

**Figure 7 nutrients-15-04399-f007:**
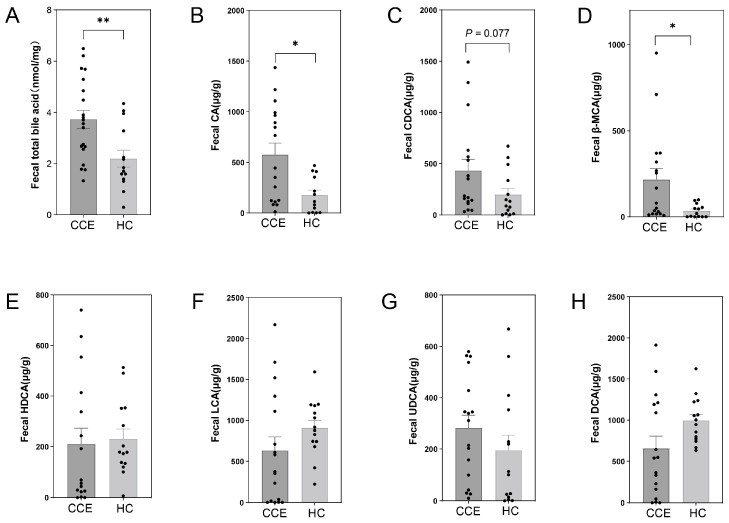
CCE changes fecal BA metabolism. (**A**) Fecal total BA content. (**B**–**H**) The content of primary and secondary BAs in the fecal samples. Fecal CA (**B**), CDCA (**C**), β-MCA (**D**), hyodeoxycholic acid (HDCA) (**E**), Lithocholic acid (LCA) (**F**), ursodeoxycholic (UDCA) (**G**), and Deoxycholic acid (DCA) (**H**) levels. * *p* ≤ 0.05, ** *p* ≤ 0.01.

**Figure 8 nutrients-15-04399-f008:**
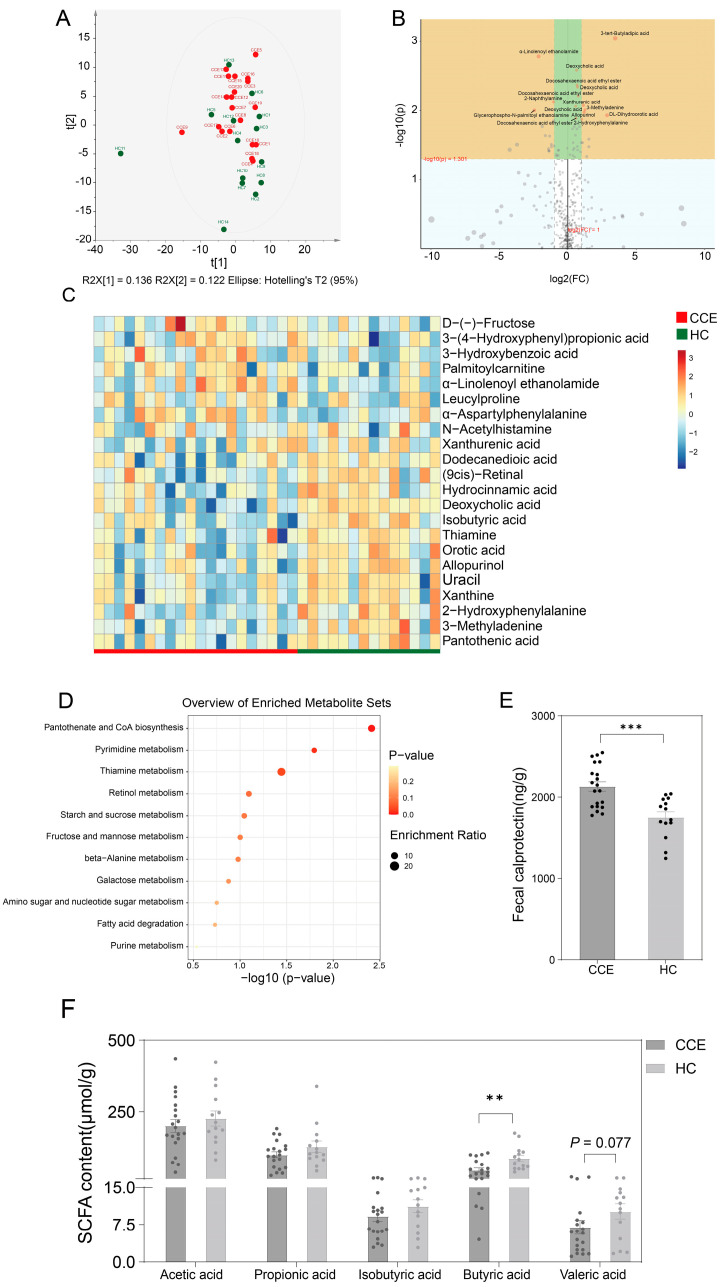
Identification of altered fecal metabolic profiles in the CCE and HC groups. (**A**) PCoA in the CCE and HC groups. (**B**) Volcanic maps showing the different metabolites between the CCE and HC groups. (**C**) Heatmap showing differential fecal metabolites (VIP > 1, *p* < 0.05) between the CCE and HC groups. The color indicates the median relative abundance of the metabolite in the group of samples. (**D**) Enrichment analysis based on KEGG metabolic pathways. (**E**) Fecal calprotectin content. (**F**) Fecal content of five SCFAs. *p* values were determined using the *t*-test and Mann-Whitney test. ** *p* ≤ 0.01, *** *p* ≤ 0.001.

**Table 1 nutrients-15-04399-t001:** Specific information from the three included studies.

Characteristic	Ren X. et al. (*n* = 104)	Li Y. et al. (*n* = 51)	Wei B. et al. (*n* = 35)
	Mean (SD)	Mean	Mean (SD)
Sex (F/M)	34/18	19/12	11/3
	34/18	12/8	15/6
Age (years)	60.02 (11.53)	46.6	60.6 (10.1)
	59.71 (11.95)	47.3	58.9 (9.9)
BMI	25.71 (3.47)	22.15	25.2 (2.3)
	24.38 (3.63)	22.1	24.8 (3.7)
Design	Case–control	Case–control	Case–control
Country	China	China	China
DNA extr.	PSP-Spin Stool Kit	QIAamp DNA Stool Minikit	TIANamp Stool DNA kit
16S region	V3-V4	V3-V4	V3-V4
Seq. Tech.	MiSeq	HiSeq	MiSeq
PE vs. SE	PE	PE	PE
Accession number	PRJNA541484	SRP247004	RJCA002279

**Table 2 nutrients-15-04399-t002:** Specific information of volunteers recruited for this study.

Characteristic	Xu F. et al. CCE (*n* = 20)	Xu F. et al. HC (*n* = 14)
Sex (F/M)	14/6	9/5
Age (years)	44.45 (10.52)	42.71 (15.09)
BMI	25.18 (2.73) *	22.58 (2.88) *
Smokers (%)	10	0
BSS	4.3 (1.1)	4.07 (0.88)
Macronutrients g/day		
Protein	73.04 (26.07)	63.04 (27.07)
Fat	75.55 (29.44) *	51.92 (20.58) *
Carbohydrates	174.69 (54.94)	149.74 (53.95)
Fiber	9.08 (4.33)	7.82 (5.31)
Cholesterol mg/day	463.39 (196.31)	413.69 (157.49)
Total energy	1695.08 (537.54) *	1326.91 (473.90) *

Values are presented as means (SD); sex difference was analyzed using the χ^2^ test and the Wilcoxon–Mann–Whitney (WMW) test for continuous data. * Significant difference compared to HC (*p* < 0.05).

## Data Availability

All data generated during and/or analyzed are available from the corresponding author on reasonable request.

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
