# Peer review of "Cholecystectomy Significantly Alters Gut Microbiota Homeostasis and Metabolic Profiles: A Cross-Sectional Study"

_nutrients, 2023, doi:10.3390/nu15204399_

Round 1
Reviewer 1 Report
In the present manuscript, Xu et al. studied the alterations in the gut microbiota after cholecystectomy (CCE), including their impact on metabolic profiles and the effects of the diet, based on both publicly available patient data and unpublished patient studies. CCE affected the composition of the gut microbiota. Analysis of the fecal metabolome indicated that CCE increased the levels of bile acids and calreticulin, with a decrease in the levels of short-chain fatty acids. These alterations are related to an increased risk of intestinal inflammation. The present study corroborates recent findings (doi: 10.3390/nu15173829), contributes insight into the side effects of CCE, and puts forward the diet as a potential mitigation strategy.
The authors should address the following minor issues:
1. For verb tense consistency, in the materials and methods (both in the main text and in the supplementary information), please describe all steps in the past tense (in some instances, there is a combination of the past and present tenses).
2. In Figure 3, please increase the size of panel D. I suggest placing panel D under panel C, maximizing panel D to the full width of Figure 3. Please also increase the resolution of panels C and D.
3. In Figure 8, please increase the width and resolution of all panels. I suggest reorganizing the figure with 4 panel rows: A+B, C, D+E, F.
4. Please include the full names of CDCA, HDCA, LCA, UDCA, and DCA.
5. In lines 356-359, the sentence "Considering the role of gut microbiota in regulating host immunity and metabolism, using the gut microbiota as a target to regulate gut microbiota homeostasis through long-term dietary or probiotic intervention strategies to maintain a stable metabolic profile may be a potential method to reduce the risk of CCE." is long and redundant (it mentions the gut microbiota three times). Please simplify it.
6. In the discussion, when summarizing the main findings of the study, please mention again the corresponding figures, in order to trace the conclusions back to the results.
7. In the discussion, please also mention the following study: doi: 10.3390/nu15173829.
8. Please include the limitations and next steps in the discussion, not in the conclusion.
9. In the supplementary materials and methods, please include a reference or website for "Compound Discoverer 3.31".
10. Please correct minor language errors, including:
10.1. Line 120: "PubMed and Web of Science was [were] used to search for publications"
10.2. Lines 129-130: "The study methods used should be consistent with [within] all articles included."
10.3. In lines 143-145, please rephrase: "Determine the nutrient content of various foods according to the China Food Composition Tables[29]. Refer to Bolte LA et al. [26]calculate the average daily nutrient intake. ".
10.4. Line 188: "GraphPad Prism 10 and R was [were] used for data analysis and visualization".
10.5. In lines 158, 166, 313, and 320, the expression "(...) in the fecal." should be completed as "in the fecal sample", "in the fecal metabolome", or equivalent.
10.6. In lines 375 and 410, please correct "between" to "among", as "between" refers to a difference between two elements only, which seems not to be the case.
10.7. In line 383, please correct the sentence "(...) can be applied used in tohe alleviate ion of various types of diseases (...)"
10.8. In Table. S2, "Exclusion studies" should be "Excluded studies". Please verify.
Please see remarks 10.1 to 10.9.
Author Response
Responds to the reviewer’s comments:
Reviewer #1:
- For verb tense consistency, in the materials and methods (both in the main text and in the supplementary information), please describe all steps in the past tense (in some instances, there is a combination of the past and present tenses).
Response: Thanks for the reviewer’s advice. Grammatical revisions have been made again to address grammatical issues. Please refer to Lines 111-112, 114-117, 118-124, 126-127, 152-153, 155-157, 180-181, 183-186, 201-205, 208-216, 250-253, 267-269, 273-274, 277, 289-293, 338-342; 2, 40, 49 (supplementary information).
Lines 111-112: On April 18, 2023, PubMed and Web of Science were used to search for publica-tions containing all keywords (…).
Lines 114-117: Subsequently, our search was refined by examining the titles of these search results, and articles that included both the terms "microbiome" (or "microbiota") and "chole-cystectomy" in their titles were considered for further assessment by reviewing their abstracts.
Lines 118-124: 1. The study compared the effects of population-based CCE on the intestinal microbiota. 2. The study had publicly available fecal raw 16S amplicon sequencing data with clearly defined subgroup information, allowing differentiation between the CCE and healthy control (HC) groups. 3. The study methods used were consistent among all selected articles. However, at the time of the search, samples from different cohort de-sign methods or geographical locations were not excluded. Among these, cohort design methods primarily included cross-sectional studies.
Lines 126-127: A total of 34 volunteers were recruited from the Wuxi Second People's Hospital of Jiangsu Province, comprising 20 individuals in the CCE group and 14 HCs.
Lines 152-153: Fecal untargeted metabolomics was analyzed using a UIUI3000 high-performance liquid chromatography (HPLC) system (Thermo Fisher Technologies (…).
Lines 155-157: Please refer to the supplementary material for detailed information on sample pre-treatment methods, HPLC-MS analytical parameters, and data analysis procedures.
Lines 180-181: Detailed fecal gut microbiota genome extraction methods are provided in the supplementary material.
Lines 183-186: In summary, raw data were analyzed using QIIME2[34]. The DADA2 package was utilized for quality filtering and demultiplexing of the raw sequencing data. Sequences with a Q-mass fraction less than 20 were filtered.
Lines 201-205: Statistical analyses were carried out using SPSS 25.0 software. Differences in microbiota characteristics, metabolic function, nutrient intake, and metabolite levels were assessed using the Wilcoxon test. The t-test was applied to analyze differential fecal un-targeted metabolites. Categorical variables were assessed using the χ2 test. Statistical significance was defined as a P-value < 0.05.
Lines 208-216: Through our search strategy, 998 studies were identified (Fig. 2). However, only three of these studies provided raw sequencing data. Consequently, four studies were included in the analysis, including those with laboratory-generated data (see Table 1). These three selected studies comprised a total of 190 samples (97 from the CCE group and 93 from the HC group), with six samples lacking annotations. This study incorpo-rated 218 samples (114 from the CCE group and 104 from the HC group). The excluded studies and their details are presented in Table S2. The characteristics of each cohort are summarized in Table 1 and 2. Given the heterogeneity observed among various studies, separate data re-analysis and combinations were conducted.
Lines 250-253: Given the notable heterogeneity observed in the gut microbiota across the four cohorts, separate tests were conducted at the family and genus levels for each cohort. This approach allowed consistent alterations in the gut microbiota to be discerned by aggregating data from the CCE and HC groups across all four cohorts.
Lines 267-269: At the family level, these mainly included Akkermansiaceae, Bacteroidaceae, Corio-bacteriaceae, Eggerthellaceae, and Muribaculaceae.
Lines 273-274: At the genus level, the AUC was 0.6942 (Fig. 4E) in the discovery cohort and 0.683 (Fig. 4F) in the validation cohort.
Line 277: Taxonomic differences were detected between the CCE and HC groups.
Lines 289-293: Further understanding of the metabolic functions associated with the gut microbiota, we conducted a KEGG pathway analysis to assess potential functional changes corresponding to taxonomic variations. Our analysis integrated data from all four studies for functional prediction. A total of 1,312 pathways were identified in both the CCE and HC groups (Fig. 5A).
Lines 338-342: Compared to the HC group, 3-(4-hydroxyphenyl) propionic acid, 3-hydroxybenzoic acid, D-(-)-fructose, leucylproline, palmitoylcarnitine, xanthurenic acid, and α-linolenoyl ethanolamide were significantly enriched in the CCE group, while xan-thine, uracil, pantothenic acid, orotic acid, and hydrocinnamic acid were significantly decreased.
Line 2: Dietary assessment and questionnaires processing
Line 40: Compound discoverers used to analyze metabolomic data
Line 49: Gas Chromatography-Mass Spectrometry (GC-MS) was used to measure the SCFA content (…).
- In Figure 3, please increase the size of panel D. I suggest placing panel D under panel C, maximizing panel D to the full width of Figure 3. Please also increase the resolution of panels C and D.
Response: Thanks for the reviewer’s advice. We have revised figure 3 to make it clearer. Please refer to Line 242.
- In Figure 8, please increase the width and resolution of all panels. I suggest reorganizing the figure with 4 panel rows: A+B, C, D+E, F.
Response: Thanks for the reviewer’s advice. We have revised Figure 8 to make it clearer. Please refer to Line 351.
- Please include the full names of CDCA, HDCA, LCA, UDCA, and DCA.
Response: Thanks for the reviewer’s advice. We have added the full name of bile acid to the manuscript. Please refer to Lines 328, 331-333.
Line 328: chenodeoxycholic acid (CDCA).
Lines 331-333: Fecal CA (B), CDCA (C), β-MCA (D), hyodeoxycholic acid (HDCA) (E), Lithocholic acid (LCA)(F), ursodeoxycholic (UDCA) (G), and Deoxycholic acid (DCA) (H) levels.
- In lines 356-359, the sentence "Considering the role of gut microbiota in regulating host immunity and metabolism, using the gut microbiota as a target to regulate gut microbiota homeostasis through long-term dietary or probiotic intervention strategies to maintain a stable metabolic profile may be a potential method to reduce the risk of CCE." is long and redundant (it mentions the gut microbiota three times). Please simplify it.
Response: Thanks for the reviewer’s advice. We have simplified the relevant expressions. Please refer to Lines 372-374.
Lines 372-374: Given the pivotal role of the gut microbiota in regulating host immunity and metabolism, strategies aimed at modulating gut microbiota balance through long-term dietary or probiotic interventions could potentially mitigate the risk of CCE.
- In the discussion, when summarizing the main findings of the study, please mention again the corresponding figures, in order to trace the conclusions back to the results.
Response: Thanks for the reviewer’s advice. We have mentioned the main result figures in the discussion section as suggested by the reviewers. Please refer to Lines 381, 382, 395, 397, 430, 448, 469, 473, 475.
- In the discussion, please also mention the following study: doi: 10.3390/nu15173829.
Response: Thanks for the reviewer’s advice. We have added relevant research (doi: 10.3390/nu15173829) to the discussion section. Please refer to Lines 369-371.
Lines 369-371: In addition, study have reported that high-fat and high-cholesterol diet promotes intestinal inflammation by exacerbating gut microbiome dysbiosis and BA disorders in CCE [38].
- 8. Please include the limitations and next steps in the discussion, not in the conclusion.
Response: Thanks for the reviewer’s advice. We have added the limitations of the study and future perspectives to the discussion section. Please refer to Lines 488-503.
Lines 488-503: Our study had several limitations. Firstly, the risk associated with post-CCE ac-cumulates over time, but we lacked precise information on the time elapsed since CCE for each sample in the cohort. Therefore, we couldn't assess the impact of CCE duration on gut microbiota homeostasis and metabolic characteristics. Secondly, the inter-action between diet and gut microbiota is complex, and we primarily focused on identifying associations between differential microbiota and dietary factors after CCE without delving into the underlying causes and potential physiological implications of this interaction. Thirdly, we used only two databases for the search, which may have resulted in missing data. Lastly, since the published studies we relied on only provided 16S data, we lacked insights into how CCE specifically alters the functional genes of the gut microbiota, such as the expression of genes encoding enzymes related to BA metabolism. For future investigations, we aspire to gain a deeper understanding of the risks linked to post-CCE and the factors that can mitigate these risks through comprehensive epidemiological studies in larger cohorts. In addition, an in-depth analysis of the alteration of gut microbiota homeostasis and related functional genes by different dietary patterns after CCE can guide us better understand the mechanism underlying the effect of CCE on intestinal health.
- In the supplementary materials and methods, please include a reference or website for "Compound Discoverer 3.31".
Response: Thanks for the reviewer’s advice. We have added relevant reference study in the supplementary materials and methods section. Please refer to Line 40 (supplementary materials).
Line 40: Compound discoverers used to analyze metabolomic data 3.31[1].
- Please correct minor language errors, including:
10.1. Line 120: "PubMed and Web of Science was [were] used to search for publications"
Response: Thanks for the reviewer’s advice. We are sorry for using the wrong expression, we have amended it. Please refer to Line 111-112.
10.2. Lines 129-130: "The study methods used should be consistent with [within] all articles included."
Response: Thanks for the reviewer’s advice. We are sorry for using the wrong expression, we have amended it. Please refer to Lines 121-122.
10.3. In lines 143-145, please rephrase: "Determine the nutrient content of various foods according to the China Food Composition Tables[29]. Refer to Bolte LA et al. [26]calculate the average daily nutrient intake. ".
Response: Thanks for the reviewer’s advice. We have revised the relevant sentence in the methods section. Please refer to Lines 146-149.
Lines 146-149: The average daily nutrient intake was calculated by multiplying frequencies of con-sumption by portion size and nutrient content per gram [26] as indicated in the China Food Composition Tables [29]. Please refer to the supplementary material for details of the FFQ.
10.4. Line 188: "GraphPad Prism 10 and R was [were] used for data analysis and visualization".
Response: Thanks for the reviewer’s advice. We are sorry for using the wrong expression, we have amended it. Please refer to Line 192.
10.5. In lines 158, 166, 313, and 320, the expression "(...) in the fecal." should be completed as "in the fecal sample", "in the fecal metabolome", or equivalent.
Response: Thanks for the reviewer’s advice. We are sorry for using the wrong expression, we have amended it. Please refer to Lines 162, 167, 175, 324.
Line 162: Briefly, pre-freeze-dried fecal samples of approximately 50 mg were weighed (…).
Line 167: Determination of BA levels in fecal samples.
Line 175: Determination of calprotectin levels in the fecal samples.
Line 324: We examined the levels of several BAs in the fecal samples.
10.6. In lines 375 and 410, please correct "between" to "among", as "between" refers to a difference between two elements only, which seems not to be the case.
Response: Thanks for the reviewer’s advice. We are sorry for using the wrong expression, we have amended it. Please refer to Lines 389-390.
Lines: 389-390: The identification of significantly altered gut microbiota after CCE is difficult owing to opposing findings among different studies
10.7. In line 383, please correct the sentence "(...) can be applied used in tohe alleviate ion of various types of diseases (...)"
Response: Thanks for the reviewer’s advice. We are sorry for using the wrong expression, we have amended it. Please refer to Lines 397-400.
Lines 397-400: Interestingly, Akkermansia [41] and Lactobacillus [42], as bacteria with positive effects on host health, can be applied to alleviate various types of diseases through a variety of pathways, such as anti-inflammatory and modulation of immune and metabolic levels.
10.8. In Table. S2, "Exclusion studies" should be "Excluded studies". Please verify.
Response: Thanks for the reviewer’s advice. We are sorry for using the wrong expression, we have amended it. Please refer to Line 72 (supplementary information).
Reviewer 2 Report
- Please add study design to the title
- Shorten the abstract and show most significant numbers (results). The abstract should attract the readers and this one is hard to follow. Make it structured too.
- Improve quality of figure 4
- Add limitation section (for instance you only used 2 databases)
Other than these comments I believe this is important manuscript and it adds to the body of literature in the field of cholecystectomy and gut microbiota. The figures, tables and references are appropriate.
Author Response
Responds to the reviewer’s comments:
Reviewer #2:
1.Please add study design to the title
Response: Thanks for the reviewer’s advice. We have added the study design to the title based on the reviewers' comments. Please refer to Lines 2-3.
Lines 2-3: Cholecystectomy significantly alters gut microbiota homeostasis and metabolic profiles:A Cross-Sectional Study.
2.Shorten the abstract and show most significant numbers (results). The abstract should attract the readers and this one is hard to follow. Make it structured too.
Response: Thanks for the reviewer’s advice. We are sorry for that our abstract showed too much detail resulting in an inability to highlight the main points of the article, which would have made it difficult for readers to understand our study. We have revised the abstract based on the main objectives, methods, results and significance of the study. Please refer to Lines 14-35.
Lines 14-35: Cholecystectomy (CCE) is a standard clinical treatment for conditions like gallstones and cholecystitis. However, its link to post-CCE syndrome, colorectal cancer, and non-alcoholic fatty liver disease has raised concerns. Additionally, studies have demonstrated the disruptive effects of CCE on gut microbiota homeostasis and bile acid (BA) metabolism. Considering the role of gut microbiota in regulating host metabolic and immune pathways, the use of dietary and probiotic intervention strategies to maintain a stable gut ecosystem after CCE could potentially reduce associated disease risks. Inter-study variations have made it challenging to identify consistent gut microbiota patterns after CCE, a prerequisite for targeted interventions. In this study, we first meta-analyzed 218 raw 16S rRNA gene sequencing data to determine consistent patterns of structural and functional changes in the gut microbiota after CCE. Our results revealed significant alterations in the gut microbiota's structure and function due to CCE. Furthermore, we identified characteristic gut microbiota changes associated with CCE by constructing a random model classifier. In the validation cohort, this classifier achieved an area under the receiver operating characteristic curve (AUC) of 0.713 and 0.683 when distinguishing between the microbiota of the CCE and healthy groups at the family and genus levels, respectively. Further, fecal metabolomics analysis demonstrated that CCE also substantially modified the metabolic profile, including decreased fecal short-chain fatty acid levels and disrupted BA metabolism. Importantly, dietary patterns, particularly excessive fat and total energy intake, influenced gut microbiota and metabolic profile changes post-CCE. These dietary habits were associated with further enrichment of the microbiota related to BA metabolism and increased levels of intestinal inflammation after CCE. In conclusion, our study identified specific alterations in gut microbiota homeostasis and metabolic pro-files associated with CCE. It also revealed a potential link between dietary patterns and gut microbiota changes following CCE. Our study provides a theoretical basis for modulating gut microbiota homeostasis after CCE using long-term dietary strategies and probiotic interventions.
3.Improve quality of figure 4
Response: Thanks for the reviewer’s advice. We have revised figure 4 to make it clearer. Please refer to Line 276.
4.Add limitation section (for instance you only used 2 databases)
Response: Thanks for the reviewer’s advice. We have added a relevant section to the discussion section based on the reviewers' comments. Please refer to Lines 495-496.
Lines 495-496: Thirdly, we used only two databases for the search, which may have resulted in missing data.

Reviewer 3 Report
Xu et al. submitted original paper entitled „Cholecystectomy significantly alters gut microbiota homeostasis and metabolic profiles: based on the interaction between food nutrients and gut microbiota”.
Authors aimed to first identify consistent patterns of the gut microbiota changes after cholecystectomy through meta-analysis and then to further analyze the effects of gut microbiota changes on metabolic profiles and the role of dietary patterns in re-shaping the microbiota after cholecystectomy through a laboratory-recruited cohort. They performed a meta-analysis of publicly available and laboratory-generated raw 16S rRNA sequencing data from studies investigating the effects of cholecystectomy on the gut microbiota. The results revealed that cholecystectomy significantly altered the structure and function of the gut microbiota. Authors identified gut microbiota characteristics altered by cholecystectomy by building a randomized model classifier. The study identified specific alterations in gut microbiota homeostasis and metabolic profiles associated with cholecystectomy and revealed a potential relationship between dietary patterns shaping the gut microbiota after cholecystectomy. This work provides a theoretical basis for modulating gut microbiota homeostasis after cholecystectomy using long-term dietary strategies and probiotic interventions. Finally, authors hope that their observations provide a new perspective for post-cholecystectomy risk assessment.
In my opinion, this paper could be interesting for the readers of Nutrients. It is well suited to the profile of the Journal. Moreover, manuscript is well prepared – either from the substantive or technical side. I have no serious comments. Authors should only check the article carefully and correct editorial errors. On the other hand, drawings require some thought. Diagrams 1 and 2 are rather schemes - not figures. The font i stoom small. Figures are illegible.
Author Response
Responds to the reviewer’s comments:
Reviewer #3:
- In my opinion, this paper could be interesting for the readers of Nutrients. It is well suited to the profile of the Journal. Moreover, manuscript is well prepared – either from the substantive or technical side. I have no serious comments. Authors should only check the article carefully and correct editorial errors. On the other hand, drawings require some thought. Diagrams 1 and 2 are rather schemes - not figures. The font i stoom small. Figures are illegible.
Response: Thanks for the reviewer’s advice. We have revised figure 3, 4, 6, and 8 to make it clearer. Please refer to Lines 242, 276, 317, 351.

Reviewer 4 Report
In the current study were identified specific alterations in gut microbiota homeostasis and metabolic profiles associated with cholecystectomy and revealed a potential relationship between dietary patterns shaping the gut microbiota after cholecystectomy. The study provides a theoretical basis for modulating gut microbiota homeostasis after cholecystectomy using long-term dietary strategies and probiotic interventions.
Some suggestions:
1. In my opinion, the abstract is too long and difficult to follow. Please shorten it by presenting only the essential information, namely what did you do in your current study.
2. The inclusion and exclusion criteria concerning the volunteer must be included in the article at point 2.1 not in the supplementary material.
3. At supplementary material you must present extensively all the informations concerning the point 2.2: Dietary assessment and processing of questionnaires.
4. The quality of Figures 3,4,6 and 8 must be improved.
5. You must add the limitations of the study at discusions not at conclusion and reformulate the conclusions.
Minor editing of English language is required
Author Response
Responds to the reviewer’s comments:
Reviewer #4:
- In my opinion, the abstract is too long and difficult to follow. Please shorten it by presenting only the essential information, namely what did you do in your current study.
Response: Thanks for the reviewer’s advice. We are sorry for that our abstract showed too much detail resulting in an inability to highlight the main points of the article, which would have made it difficult for readers to understand our study. We have revised the abstract based on the main objectives, methods, results and significance of the study. Please refer to Lines 14-35.
Lines 14-35: Cholecystectomy (CCE) is a standard clinical treatment for conditions like gallstones and cholecystitis. However, its link to post-CCE syndrome, colorectal cancer, and non-alcoholic fatty liver disease has raised concerns. Additionally, studies have demonstrated the disruptive effects of CCE on gut microbiota homeostasis and bile acid (BA) metabolism. Considering the role of gut microbiota in regulating host metabolic and immune pathways, the use of dietary and probiotic intervention strategies to maintain a stable gut ecosystem after CCE could potentially reduce associated disease risks. Inter-study variations have made it challenging to identify consistent gut microbiota patterns after CCE, a prerequisite for targeted interventions. In this study, we first meta-analyzed 218 raw 16S rRNA gene sequencing data to determine consistent patterns of structural and functional changes in the gut microbiota after CCE. Our results revealed significant alterations in the gut microbiota's structure and function due to CCE. Furthermore, we identified characteristic gut microbiota changes associated with CCE by constructing a random model classifier. In the validation cohort, this classifier achieved an area under the receiver operating characteristic curve (AUC) of 0.713 and 0.683 when distinguishing between the microbiota of the CCE and healthy groups at the family and genus levels, respectively. Further, fecal metabolomics analysis demonstrated that CCE also substantially modified the metabolic profile, including decreased fecal short-chain fatty acid levels and disrupted BA metabolism. Importantly, dietary patterns, particularly excessive fat and total energy intake, influenced gut microbiota and metabolic profile changes post-CCE. These dietary habits were associated with further enrichment of the microbiota related to BA metabolism and increased levels of intestinal inflammation after CCE. In conclusion, our study identified specific alterations in gut microbiota homeostasis and metabolic pro-files associated with CCE. It also revealed a potential link between dietary patterns and gut microbiota changes following CCE. Our study provides a theoretical basis for modulating gut microbiota homeostasis after CCE using long-term dietary strategies and probiotic interventions.
- The inclusion and exclusion criteria concerning the volunteer must be included in the article at point 2.1 not in the supplementary material.
Response: Thanks for the reviewer’s advice. We have added inclusion and exclusion criteria for volunteers to point 2.1. Please refer to Lines 131-142.
Lines 131-142:
Inclusion and exclusion criteria
Patients in the CCE group were selected based on the following criteria: (1) CCE performed more than 2 years ago; (2) age between 18 and 65 years; and (3) provision of written informed consent. Patients were excluded if they had (1) a surgical history of gastrointestinal tract procedures; (2) a medical history of irritable bowel syndrome, IBD, constipation, or infective or idiopathic diarrhea; (3) a medication history of anti-biotics, probiotics, or drugs known to affect gut microbiota within the past month; or (4) a history of severe chronic diseases.
HCs voluntarily participated in the study and were selected based on the following criteria: (1) age between 18 and 65 years; (2) provision of written informed consent; (3) no history of GB removal surgery or other gastrointestinal surgeries; and (4) no administration of antibiotics or probiotics in the month preceding the study.
- At supplementary material you must present extensively all the informations concerning the point 2.2: Dietary assessment and processing of questionnaires.
Response: Thanks for the reviewer’s advice. We have added the details of the FFQ to the supplementary material. Please refer to Lines 2-6 of the supplementary material.
Lines 2-6: Dietary intake assessment included whether the food was consumed, consumption frequency (times of per day/week/month/year) and the average amount of food consumption at each time. The 149 food items in the FFQ were classifed into 18 predefned food groups based on similarities in nutrient profle and culinary usage.
- The quality of Figures 3,4,6 and 8 must be improved.
Response: Thanks for the reviewer’s advice. We have revised figure 3, 4, 6, and 8 to make it clearer. Please refer to Lines 242, 276, 317, 351.
- You must add the limitations of the study at discusions not at conclusion and reformulate the conclusions.
Response: Thanks for the reviewer’s advice. We have added the limitations of the study and future perspectives to the discussion section. In addition, we further summarize the findings in the conclusions section. Please refer to Lines 488-503, 505-518.
Lines 488-503: Our study had several limitations. Firstly, the risk associated with post-CCE ac-cumulates over time, but we lacked precise information on the time elapsed since CCE for each sample in the cohort. Therefore, we couldn't assess the impact of CCE duration on gut microbiota homeostasis and metabolic characteristics. Secondly, the inter-action between diet and gut microbiota is complex, and we primarily focused on identifying associations between differential microbiota and dietary factors after CCE without delving into the underlying causes and potential physiological implications of this interaction. Thirdly, we used only two databases for the search, which may have resulted in missing data. Lastly, since the published studies we relied on only provided 16S data, we lacked insights into how CCE specifically alters the functional genes of the gut microbiota, such as the expression of genes encoding enzymes related to BA metabolism. For future investigations, we aspire to gain a deeper understanding of the risks linked to post-CCE and the factors that can mitigate these risks through comprehensive epidemiological studies in larger cohorts. In addition, an in-depth analysis of the alteration of gut microbiota homeostasis and related functional genes by different dietary patterns after CCE can guide us better understand the mechanism underlying the effect of CCE on intestinal health.
Lines 505-518: Our study revealed that CCE significantly altered gut microbiota homeostasis and associated metabolic profiles. On this basis, we further demonstrated the role of different dietary patterns in remodeling gut microbiota homeostasis after CCE. Enrichment of microbiota associated with BA metabolism and altered function associated with lipid metabolism are key features accompanied by elevated fecal calprotectin and total BA levels and decreased SCFAs content. Collectively, these findings imply a heightened risk of intestinal inflammation associated with CCE and underscore the pivotal role played by the disruption of gut microbiota equilibrium and metabolic pro-files in the development of intestinal inflammation. This study presents a fresh perspective for assessing the risks following CCE, moving beyond mere correlations and delving into potential causative factors. Furthermore, it lays the foundation for targeted interventions aimed at enhancing post-CCE intestinal health. Such interventions may involve long-term dietary strategies and prebiotic interventions, harnessing the power of gut microbiota modulation to mitigate post-CCE risks and improve patient outcomes.
- Comments on the Quality of English Language
Minor editing of English language is required
Response: Thanks for the reviewer’s advice. We have further checked the English presentation issues and have amended them accordingly. Please refer to Lines 111-112, 114-117, 118-124, 126-127, 152-153, 155-157, 180-181, 183-186, 201-205, 208-216, 250-253, 267-269, 273-274, 277, 289-293, 338-342.
Lines 111-112: On April 18, 2023, PubMed and Web of Science were used to search for publica-tions containing all keywords (…).
Lines 114-117: Subsequently, our search was refined by examining the titles of these search results, and articles that included both the terms "microbiome" (or "microbiota") and "chole-cystectomy" in their titles were considered for further assessment by reviewing their abstracts.
Lines 118-124: 1. The study compared the effects of population-based CCE on the intestinal microbiota. 2. The study had publicly available fecal raw 16S amplicon sequencing data with clearly defined subgroup information, allowing differentiation between the CCE and healthy control (HC) groups. 3. The study methods used were consistent among all selected articles. However, at the time of the search, samples from different cohort de-sign methods or geographical locations were not excluded. Among these, cohort design methods primarily included cross-sectional studies.
Lines 126-127: A total of 34 volunteers were recruited from the Wuxi Second People's Hospital of Jiangsu Province, comprising 20 individuals in the CCE group and 14 HCs.
Lines 152-153: Fecal untargeted metabolomics was analyzed using a UIUI3000 high-performance liquid chromatography (HPLC) system (Thermo Fisher Technologies (…).
Lines 155-157: Please refer to the supplementary material for detailed information on sample pre-treatment methods, HPLC-MS analytical parameters, and data analysis procedures.
Lines 180-181: Detailed fecal gut microbiota genome extraction methods are provided in the supplementary material.
Lines 183-186: In summary, raw data were analyzed using QIIME2[34]. The DADA2 package was utilized for quality filtering and demultiplexing of the raw sequencing data. Sequences with a Q-mass fraction less than 20 were filtered.
Lines 201-205: Statistical analyses were carried out using SPSS 25.0 software. Differences in microbiota characteristics, metabolic function, nutrient intake, and metabolite levels were assessed using the Wilcoxon test. The t-test was applied to analyze differential fecal un-targeted metabolites. Categorical variables were assessed using the χ2 test. Statistical significance was defined as a P-value < 0.05.
Lines 208-216: Through our search strategy, 998 studies were identified (Fig. 2). However, only three of these studies provided raw sequencing data. Consequently, four studies were included in the analysis, including those with laboratory-generated data (see Table 1). These three selected studies comprised a total of 190 samples (97 from the CCE group and 93 from the HC group), with six samples lacking annotations. This study incorpo-rated 218 samples (114 from the CCE group and 104 from the HC group). The excluded studies and their details are presented in Table S2. The characteristics of each cohort are summarized in Table 1 and 2. Given the heterogeneity observed among various studies, separate data re-analysis and combinations were conducted.
Lines 250-253: Given the notable heterogeneity observed in the gut microbiota across the four cohorts, separate tests were conducted at the family and genus levels for each cohort. This approach allowed consistent alterations in the gut microbiota to be discerned by aggregating data from the CCE and HC groups across all four cohorts.
Lines 267-269: At the family level, these mainly included Akkermansiaceae, Bacteroidaceae, Corio-bacteriaceae, Eggerthellaceae, and Muribaculaceae.
Lines 273-274: At the genus level, the AUC was 0.6942 (Fig. 4E) in the discovery cohort and 0.683 (Fig. 4F) in the validation cohort.
Line 277: Taxonomic differences were detected between the CCE and HC groups.
Lines 289-293: Further understanding of the metabolic functions associated with the gut microbiota, we conducted a KEGG pathway analysis to assess potential functional changes corresponding to taxonomic variations. Our analysis integrated data from all four studies for functional prediction. A total of 1,312 pathways were identified in both the CCE and HC groups (Fig. 5A).
Lines 338-342: Compared to the HC group, 3-(4-hydroxyphenyl) propionic acid, 3-hydroxybenzoic acid, D-(-)-fructose, leucylproline, palmitoylcarnitine, xanthurenic acid, and α-linolenoyl ethanolamide were significantly enriched in the CCE group, while xan-thine, uracil, pantothenic acid, orotic acid, and hydrocinnamic acid were significantly decreased.
Line 2: Dietary assessment and questionnaires processing
Line 40: Compound discoverers used to analyze metabolomic data
Line 49: Gas Chromatography-Mass Spectrometry (GC-MS) was used to measure the SCFA content (…).

Round 2
Reviewer 4 Report
The manuscript has been significantly improved but unfortunately the quality of the figures was not improved.
Minor editing of English language is required.